# Meta Flow Matching:
# Integrating Vector Fields on the Wasserstein Manifold

## Abstract

Numerous biological and physical processes can be modeled as systems of interact-
ing samples evolving continuously over time, e.g. the dynamics of communicating
cells or physical particles. Flow-based models allow for learning these dynamics at
the population level — they model the evolution of the entire distribution of sam-
ples. However, current flow-based models are limited to a single initial population
and a set of predefined conditions which describe different dynamics. We argue that
multiple processes in natural sciences have to be represented as vector fields on the
Wasserstein manifold of probability densities. That is, the change of the population
at any moment in time depends on the population itself due to the interactions
between samples. In particular, this is crucial for personalized medicine where the
development of diseases and their treatments depend on the microenvironment of
cells specific to each patient. We propose *Meta Flow Matching* (MFM), a practical
approach to integrating along these vector fields on the Wasserstein manifold by
amortizing the flow model over the initial populations. Namely, we embed the
population of samples using a Graph Neural Network (GNN) and use these embed-
dings to train a *Flow Matching* model. This gives Meta Flow Matching the ability
to generalize over the initial distributions unlike previously proposed methods.
Finally, we demonstrate the ability of MFM to improve prediction of individual
treatment responses on a large scale multi-patient single-cell drug screen dataset.

## 1 Introduction

Understanding the dynamics of many-body problems is a central challenge across the natural sciences.
In the field of cell biology, a central focus is the understanding of the dynamic processes that cells
undergo in response to their environment, and in particular their response and interaction with other
cells. Cells communicate with one other in close proximity using *cell signaling*, exerting influence
over each other's trajectories (Armingol et al., 2020; Goodenough and Paul, 2009). This signaling
presents an obstacle for modeling, but is essential for understanding and eventually controlling
cell dynamics during development (Gulati et al., 2020; Rizvi et al., 2017), in diseased states (Molè
et al., 2021; Binnewies et al., 2018; Zeng and Dai, 2019; Chung et al., 2017), and in response to
perturbations (Ji et al., 2021; Peidli et al., 2024).

The super-exponential decrease of sequencing costs and advances in microfluidics has enabled the
rapid advancement of single-cell sequencing and related technologies over the past decade. While
single-cell sequencing has been used to great effect to understand the heterogeneity in cell systems,
they are also destructive, making longitudinal measurements extremely difficult. Instead, most
approaches model cell dynamics at the population level (Hashimoto et al., 2016; Weinreb et al., 2018;
Schiebinger et al., 2019; Tong et al., 2020; Neklyudov et al., 2022; Bunne et al., 2023a). These
approaches involve the formalisms of optimal transport (Villani, 2009; Peyré and Cuturi, 2019) and

Submitted to 38th Conference on Neural Information Processing Systems (NeurIPS 2024). Do not distribute.

generative modeling (De Bortoli et al., 2021; Lipman et al., 2023) methods, which allow for learning a map between empirical measures. While these methods are able to model the dynamics of the population, they are fundamentally limited in that they model the evolution of cells as independent particles evolving according to a shared dynamical system. Furthermore, these models can be trained to match any given set of measures, but they are restricted to modeling of a single population and can at best condition on a number of different dynamics that is available in the training data.

To address this we propose *Meta Flow Matching* (MFM) — the amortization of the Flow Matching generative modeling framework (Lipman et al., 2023) over the input measures. In practice, our method can be used to predict the time-evolution of distributions from a given dataset of the time-evolved examples. Namely, we assume that the collected data undergoes a universal developmental process, which depends only on the population itself as in the setting of the interacting particles or communicating cells. Under this assumption, we learn the vector field model that takes samples from the initial distribution as input and defines the push-forward map on the sample-space that maps the initial distribution to the final distribution.

We showcase the utility of our approach on two applications. We first explore Meta Flow Matching on a synthetic task of denoising letters. We show that MFM is able to generalize the denoising process to letters in unseen orientations where a standard flow matching approach cannot. Next, we explore how MFM can be applied to model single-cell perturbation data (Ji et al., 2021; Peidli et al., 2024). We evaluate MFM on predicting the response of patient-derived cells to chemotherapy treatments in a recently published large scale single-cell drug screening dataset where there are known to be patient-specific responses (Ramos Zapatero et al., 2023). This dataset includes more than 25M cells collected over ten patients under 2500 conditions. This is a challenging task due to the variance over multiple patients, treatments applied and the local cell compositions, but it can be used to study the *tumor micro-environment* (TME), thought to be essential in circumventing chemoresistance. We demonstrate that Meta Flow Matching can successfully predict the development of cell populations on replicated experiments, and, most importantly, it generalizes to previously unseen patients, thus, capturing the patient-specific response to the treatment.

## 2  Background

### 2.1  Generative Modeling via Flow Matching

Flow Matching is an approach to generative modeling recently proposed independently in different works: Rectified Flows (Liu et al., 2022), Flow Matching (Lipman et al., 2023), Stochastic Interpolants (Albergo and Vanden-Eijnden, 2022). It assumes a continuous interpolation between densities $p_0(x_0)$ and $p_1(x_1)$ in the sample space. That is, the sample from the intermediate density $p_t(x_t)$ is produced as follows

$$x_t = f_t(x_0, x_1), \ \ (x_0, x_1) \sim \pi(x_0, x_1), \tag{1}$$

$$\text{where} \int dx_1 \ \pi(x_0, x_1) = p_0(x_0), \ \ \int dx_0 \ \pi(x_0, x_1) = p_1(x_1), \tag{2}$$

where $f_t$ is the time-continuous interpolating function such that $f_{t=0}(x_0, x_1) = x_0$ and $f_{t=1}(x_0, x_1) = x_1$ (e.g. linearly between $x_0$ and $x_1$ with $f_t(x_0, x_1) = (1 - t) \cdot x_0 + t \cdot x_1$); $\pi(x_0, x_1)$ is the density of the joint distribution, which is usually taken as a distribution of independent random variables $\pi(x_0, x_1) = p_0(x_0)p_1(x_1)$, but can also be generalized to formulate the optimal transport problems (Pooladian et al., 2023; Tong et al., 2024). The corresponding density can be defined then as the following expectation

$$p_t(x) = \int dx_0 dx_1 \ \pi(x_0, x_1)\delta(x - f_t(x_0, x_1)). \tag{3}$$

The essential part of Flow Matching is the continuity equation that describes the change of this density through the vector field on the state space, which admits vector field $v_t^*(x)$ as a solution

$$\frac{\partial p_t(x)}{\partial t} = -\langle \nabla_x, p_t(x)v_t^*(x)\rangle, \ \ v_t^*(\xi) = \frac{1}{p_t(\xi)}\mathbb{E}_{\pi(x_0,x_1)}\left[\delta(f_t(x_0, x_1) - \xi)\frac{\partial f_t(x_0, x_1)}{\partial t}\right]. \tag{4}$$

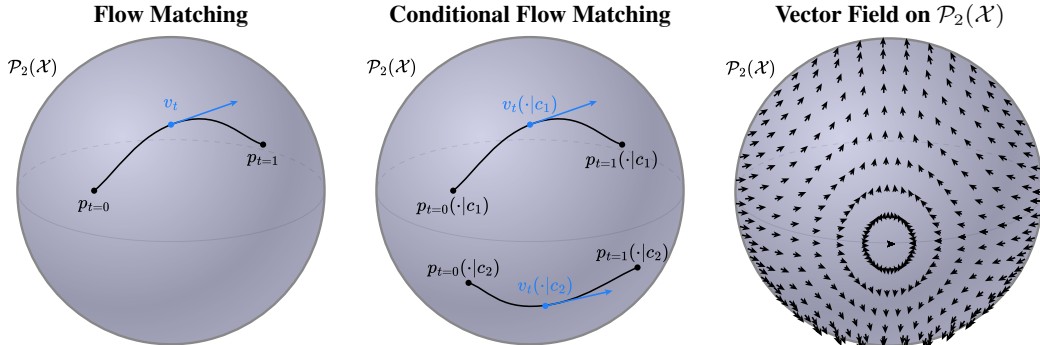

Figure 1: Illustration of flow matching methods on the 2-Wasserstein manifold, $\mathcal{P}_2(\mathcal{X})$, depicted as a two-dimensional sphere. *Flow Matching* learns the tangent vectors to a single curve on the manifold. *Conditional* generation corresponds to learning a finite set of curves on the manifold, e.g. classes $c_1$ and $c_2$ on the plot. *Meta Flow Matching* learns to integrate a vector field on $\mathcal{P}_2(\mathcal{X})$, i.e. for every starting density $p_0$ Meta Flow Matching defines a push-forward measure that integrates along the underlying vector field.

Relying on this formula, one can derive the tractable objective for learning $v_t^*(x)$, i.e.

$$\mathcal{L}_{\text{FM}}(\omega) = \int_0^1 dt \, \mathbb{E}_{p_t(x)} \|v_t^*(x) - v_t(x; \omega)\|^2 \tag{5}$$

$$= \mathbb{E}_{\pi(x_0, x_1)} \int_0^1 dt \left\| \frac{\partial}{\partial t} f_t(x_0, x_1) - v_t(f_t(x_0, x_1); \omega) \right\|^2 + \text{constant}. \tag{6}$$

Finally, the vector field $v_t(\xi, \omega) \approx v_t^*(\xi)$ defines the push-forward density that approximately matches $p_{t=1}$, i.e. $T_\# p_0 \approx p_{t=1}$, where $T$ is the flow corresponding to vector field $v_t(\cdot, \omega)$ with parameters $\omega$.

## 2.2 Conditional Generative Modeling via Flow Matching

Conditional image generation is one of the most common applications of generative models nowadays; it includes conditioning on the text prompts (Saharia et al., 2022b; Rombach et al., 2022) as well as conditioning on other images (Saharia et al., 2022a). To learn the conditional generative process with diffusion models, one merely has to pass the conditional variable (sampled jointly with the data point) as an additional input to the parametric model of the vector field. The same applies for the Flow Matching framework.

Conditional Generative Modeling via Flow Matching is independently introduced in several works (Zheng et al., 2023; Dao et al., 2023; Isobe et al., 2024) and it operates as follows. Consider a family of time-continuous densities $p_t(x_t \,|\, c)$, which corresponds to the distribution of the following random variable

$$x_t = f_t(x_0, x_1), \quad (x_0, x_1) \sim \pi(x_0, x_1 \,|\, c). \tag{7}$$

For every $c$, the density $p_t(x_t \,|\, c)$ follows the continuity equation with the following vector field

$$v_t^*(\xi \,|\, c) = \frac{1}{p_t(\xi \,|\, c)} \mathbb{E}_{\pi(x_0, x_1)} \delta(f_t(x_0, x_1) - \xi) \frac{\partial f_t(x_0, x_1)}{\partial t}, \tag{8}$$

which depends on $c$. Thus, the training objective of the conditional model becomes

$$\mathcal{L}_{CGFM}(\omega) = \mathbb{E}_{p(c)} \mathbb{E}_{\pi(x_0, x_1 \,|\, c)} \int_0^1 dt \left\| \frac{\partial}{\partial t} f_t(x_0, x_1) - v_t(f_t(x_0, x_1) \,|\, c; \omega) \right\|^2, \tag{9}$$

where, compared to the original Flow Matching formulation, we first have to sample $c$, then produce the samples from $p_t(x_t \,|\, c)$ and pass $c$ as input to the parametric model of the vector field.

## 3 Meta Flow Matching

In this paper, we propose the amortization of the Flow Matching framework over the marginal distributions. Our model is based on the outstanding ability of the Flow Matching framework to

learn the push-forward map for any joint distribution $\pi(x_0, x_1)$ given empirically. For the given joint $\pi(x_0, x_1)$, we denote the solution of the Flow Matching optimization problem as follows

$$v_t^*(\cdot, \pi) = \underset{v_t}{\arg\min}\, \mathcal{L}_{GFM}(v_t(\cdot), \pi(x_0, x_1))\,. \tag{10}$$

Analogously to the amortized optimization (Chen et al., 2022; Amos et al., 2023), we aim to learn the model that outputs the solution of Eq. (10) based on the input data sampled from $\pi$, i.e.

$$v_t(\cdot, \varphi(\pi)) = v_t^*(\cdot, \pi)\,, \tag{11}$$

where $\varphi(\pi)$ is the embedding model of $\pi$ and the joint density $\pi(\cdot \,|\, c)$ is generated using some unknown measure of the conditional variables $c \sim p(c)$.

## 3.1 Modeling Process in Natural Sciences as Vector Fields on the Wasserstein Manifold

We argue that numerous biological and physical processes cannot be modeled via the vector field propagating the population samples independently. Thus, we propose to model these processes as families of conditional vector fields where we amortize the conditional variable by embedding the population via a Graph Neural Network (GNN).

To provide the reader with the necessary intuition, we are going to use the geometric formalism developed by Otto (2001). That is, time-dependent densities $p_t(x_t)$ define absolutely-continuous curves on the 2-Wasserstein space of distributions $\mathcal{P}_2(\mathcal{X})$ (Ambrosio et al., 2008). The tangent space of this manifold is defined by the gradient flows $\mathcal{S}_t = \{\nabla s_t \,|\, s_t : \mathcal{X} \to \mathbb{R}\}$ on the state space $\mathcal{X}$. In the Flow Matching context, we are going to refer to the tangent vectors as vector fields since one can always project the vector field onto the tangent space by parameterizing it as a gradient flow (Neklyudov et al., 2022).

Under the geometric formalism of the 2-Wasserstein manifold, Flow Matching can be considered as learning the tangent vectors $v_t(\cdot)$ along the density curve $p_t(x_t)$ defined by the sampling process in Eq. (2) (see the left panel in Fig. 1). Furthermore, the conditional generation processes $p_t(x_t \,|\, c)$ would be represented as a finite set of curves if $c$ is discrete (e.g. class-conditional generation of images) or as a family of curves if $c$ is continuous (see the middle panel in Fig. 1).

Finally, one can define a vector field on the 2-Wasserstein manifold via the continuity equation with the vector field $v_t(x, p_t(x))$ on the state space $\mathcal{X}$ that depends on the current density $p_t(x)$ or its derivatives. Below we give two examples of processes defined as vector fields on the 2-Wasserstein manifold.

**Example 1** (Mean-field limit of interacting particles). *In the limit of the infinite number of interacting particles one can describe their state with the density function $p_t(x)$. Consider the interaction according to the first order dynamics with the velocity $k(x, y) : \mathbb{R}^d \times \mathbb{R}^d \to \mathbb{R}^d$ of the particles at point $x$ that interact with the particles at point $y$. Then the change of the density is described by the following continuity equation*

$$\frac{dx}{dt} = \mathbb{E}_{p_t(y)} k(x, y), \quad \frac{\partial p_t(x)}{\partial t} = -\big\langle \nabla_x, p_t(x) \mathbb{E}_{p_t(y)} k(x, y) \big\rangle\,. \tag{12}$$

**Example 2** (Diffusion). *Even when the physical particles evolve independently in nature, the deterministic vector field model might be dependent on the current density of the population. For instance, for the diffusion process, the change of the density is described by the Fokker-Planck equation, which results in the density-dependent vector field when written as a continuity equation, i.e.*

$$\frac{\partial p_t(x)}{\partial t} = \frac{1}{2}\Delta_x p_t(x) = -\Big\langle \nabla_x, p_t(x)\Big(-\frac{1}{2}\nabla_x \log p_t(x)\Big)\Big\rangle \implies \frac{dx}{dt} = -\frac{1}{2}\nabla_x \log p_t(x)\,. \tag{13}$$

Motivated by the examples above, we argue that using the information about the current or the initial density is crucial for the modeling of time-evolution of densities in natural processes, to capture this type of dependency one can model the change of the density as the following Cauchy problem

$$\frac{\partial p_t(x)}{\partial t} = -\big\langle \nabla_x, p_t(x) v_t(x, p_t)\big\rangle\,, \quad p_{t=0}(x) = p_0(x)\,, \tag{14}$$

where the state-space vector field $v_t(x, p_t)$ depends on the density $p_t$.

The dependency might vary across models, e.g. in Example 1 the vector field can be modeled as an application of a kernel to the density function, while in Example 2 the vector field depends only on the local value of the density and its derivative.

## 3.2 Integrating Vector Fields on the Wasserstein Manifold via Meta Flow Matching

Consider the dataset of joint populations $\mathcal{D} = \{(\pi(x_0, x_1 \,|\, i))\}_i$, where, to simplify the notation, we associate every $i$-th population with its density $\pi(\cdot \,|\, i)$ and the conditioning variable here is the index of this population in the dataset. We make the following assumptions regarding the ground truth sampling process (i) we assume that the starting marginals $p_0(x_0 \,|\, i) = \int dx_1 \, \pi(x_0, x_1 \,|\, i)$ are sampled from some unknown distribution that can be parameterized with a large enough number of parameters (ii) the endpoint marginals $p_1(x_1 \,|\, i) = \int dx_0 \, \pi(x_0, x_1 \,|\, i)$ are obtained as push-forward densities solving the Cauchy problem in Eq. (14), (iii) there exists unique solution to this Cauchy problem.

One can learn a joint model of all the processes from the dataset $\mathcal{D}$ using the conditional version of the Flow Matching algorithm (see Section 2.2) where the population index $i$ plays the role of the conditional variable. However, obviously, such a model will not generalize beyond the considered data $\mathcal{D}$ and unseen indices $i$. We illustrate this empirically in Section 5.

To be able to generalize to previously unseen populations, we propose learning the density-dependent vector field motivated by Eq. (14). That is, we propose to use an embedding function $\varphi : \mathcal{P}_2(\mathcal{X}) \rightarrow \mathbb{R}^m$ to embed the starting marginal density $p_0$, which we then input into the vector field model and minimize the following objective over $\omega$

$$\mathcal{L}_{\text{MFM}}(\omega; \varphi) = \mathbb{E}_{i \sim \mathcal{D}} \mathbb{E}_{\pi(x_0, x_1 \,|\, i)} \int_0^1 dt \left\| \frac{\partial}{\partial t} f_t(x_0, x_1) - v_t(f_t(x_0, x_1) \,|\, \varphi(p_0); \omega) \right\|^2. \tag{15}$$

Note that the initial density $p_0$ is enough to predict the push-forward density $p_1$ since the Cauchy problem for Eq. (14) has a unique solution. The embedding function $\varphi(p_0)$ can take different forms, e.g. it can be the density value $\varphi(p_0) = p_0(\cdot)$, which is then used inside the vector field model to evaluate at the current point (analogous to Example 2); a kernel density estimator (analogous to Example 1); or a parametric model taking the samples from this density as an input.

**Proposition 1.** *Meta Flow Matching recovers the Conditional Generation via Flow Matching when the conditional dependence of the marginals $p_0(x_0 \,|\, c) = \int dx_1 \pi(x_0, x_1 \,|\, c)$ and $p_1(x_1 \,|\, c) = \int dx_0 \pi(x_0, x_1 \,|\, c)$ and the distribution $p(c)$ are known, i.e. there exist $\varphi : \mathcal{P}_2(\mathcal{X}) \rightarrow \mathbb{R}^m$ such that $\mathcal{L}_{MFM}(\omega) = \mathcal{L}_{CGFM}(\omega)$.*

*Proof.* Indeed, sampling from the dataset $i \sim \mathcal{D}$ becomes sampling of the conditional variable $c \sim p(c)$ and the embedding function becomes $\varphi(p_0(\cdot \,|\, c)) = c$. $\qquad\square$

Furthermore, for the parametric family of the embedding models $\varphi(p_t, \theta)$, we show that the parameters $\theta$ can be estimated by minimizing the objective in Eq. (15) in the joint optimization with the vector field parameters $\omega$. We formalize this statement in the following theorem.

**Theorem 1.** *Consider a dataset of populations $\mathcal{D} = \{(\pi(x_0, x_1 \,|\, i))\}_i$ generated from some unknown conditional model $\pi(x_0, x_1 \,|\, c)p(c)$. Then the following objective*

$$\mathcal{L}(\omega, \theta) = \mathbb{E}_{p(c)} \int_0^1 dt \, \mathbb{E}_{p_t(x_t \,|\, c)} \| v_t^*(x_t \,|\, c) - v_t(x_t \,|\, \varphi(p_0, \theta), \omega) \|^2 \tag{16}$$

*is equivalent to the Meta Flow Matching objective*

$$\mathcal{L}_{MFM}(\omega, \theta) = \mathbb{E}_{i \sim \mathcal{D}} \mathbb{E}_{\pi(x_0, x_1 \,|\, i)} \int_0^1 dt \left\| \frac{\partial}{\partial t} f_t(x_0, x_1) - v_t(f_t(x_0, x_1) \,|\, \varphi(p_0, \theta); \omega) \right\|^2 \tag{17}$$

*up to an additive constant.*

*Proof.* We postpone the proof to Appendix A. $\qquad\square$

## 3.3 Learning Population Embeddings via Graph Neural Networks (GNNs)

In many applications, the populations $\mathcal{D} = \{(\pi(x_0, x_1 \,|\, i))\}_{i=1}^N$ are given as empirical distributions, i.e. they are represented as samples from some unknown density $\pi$

$$\{(x_0^j, x_1^j)\}_{j=1}^{N_i}, \quad (x_0^j, x_1^j) \sim \pi(x_0, x_1 \,|\, i), \tag{18}$$

where $N_i$ is the size of the $i$-th population. For instance, for the diffusion process considered in Example 2, the samples from $\pi(x_0, x_1 \mid i)$ can be generated by generating some marginal $p_1(x_1 \mid i)$ and then adding the Gaussian random variable to the samples $x_1^j$. We use this model in our synthetic experiments in Section 5.1.

Since the only available information about the populations is samples, we propose learning the embedding of populations via a parametric model $\varphi(p_0, \theta)$, i.e.

$$\varphi(p_0, \theta) = \varphi\left(\{x_0^j\}_{j=1}^{N_i}, \theta\right), \quad (x_0^j, x_1^j) \sim \pi(x_0, x_1 \mid i). \tag{19}$$

For this purpose, we employ GNNs, which recently have been successfully applied for simulation of complicated many-body problems in physics (Sanchez-Gonzalez et al., 2020). To embed a population $\{x_0^j\}_{j=1}^{N_i}$, we create a k-nearest neighbour graph $G_i$ based on the metric in the state-space $\mathcal{X}$, input it into a GNN, which consists of several message-passing iterations (Gilmer et al., 2017) and the final average-pooling across nodes to produce the embedding vector. Finally, we update the parameters of the GNN jointly with the parameters of the vector field to minimize the loss function in Eq. (17).

# 4 Related Work

The meta-learning of probability measures was previously studied by Amos et al. (2022) where they demonstrate that the prediction of the optimal transport paths can be efficiently amortized over the input marginal measures. The main difference with our approach is that we are trying to learn the push-forward map without embedding the second marginal.

**Generative modeling for single cells.** Single cell data has expanded to encompass multiple modalities of data profiling cell state and activities (Frangieh et al., 2021; Bunne et al., 2023b). Single-cell data presents multiple challenges in terms of noise, non-time resolved, and high dimension, and generative models have been used to counter those problems. Autoencoder has been used to embed and extrapolate data Out Of Distribution (OOD) with its latent state dimension (Lotfollahi et al., 2019; Lopez et al., 2018; Hetzel et al., 2022). Orthogonal non-negative matrix factorization (oNMF) has also been used for dimensionality reduction combined with mixture models for cell state prediction (Chen et al., 2020). Other approaches have tried to use Flow Matching (FM) (Tong et al., 2023, 2024; Neklyudov et al., 2023) or similar approaches such as the Monge gap (Uscidda and Cuturi, 2023) to predict cell trajectories. Currently, the state of the art method uses the principle of Optimal Transport (OT) to predict cell trajectories with Input Convex Neural Network (ICNN) (Makkuva et al., 2020; Bunne et al., 2023b). What determines the significance of the method is its capability in generalizing out of distribution to a new population of cells, which may be from different culture or individuals. As of this time, our method is the only method that takes inter-cellular interactions into account.

**Generative modeling for physical processes.** The closest approach to ours is the prediction of the many-body interactions in physics (Sanchez-Gonzalez et al., 2020) via GNNs. However, the problem there is very different since these models use the information about the individual trajectories of samples, which are not available for the single-cell prediction. Neklyudov et al. (2022) consider learning the vector field for any continuous time-evolution of a probability measure, however, their method is restricted to single curves and do not consider generalization to unseen data. Finally, the weather/climate forecast models generating the next state conditioned on the previous one (Price et al., 2023; Verma et al., 2024) are similar approaches to ours but operating on a much finer time resolution.

# 5 Experiments

To show the effectiveness of MFM to generalize under previously unseen populations for the task population prediction, we consider two experimental settings. (i) A synthetic experiment with well defined coupled populations, and (ii) experiments on a publicly available single-cell dataset consisting of populations from patient dependent treatment response trials. To quantify model performance, we consider three distributional distances metrics: the 1-Wasserstein distance ($\mathcal{W}_1$), 2-Wasserstein ($\mathcal{W}_2$) distance, and the radial basis kernel maximum-mean-discrepancy (MMD) distance (Gretton et al., 2012). We parameterize all vector field models $v_t(\cdot \mid \varphi(p_0); \omega)$ using a Multi-Layer Perceptron (MLP). For MFM, we additionally parameterize $\varphi(p_t; \theta, k)$ using a Graph Convolutional Network

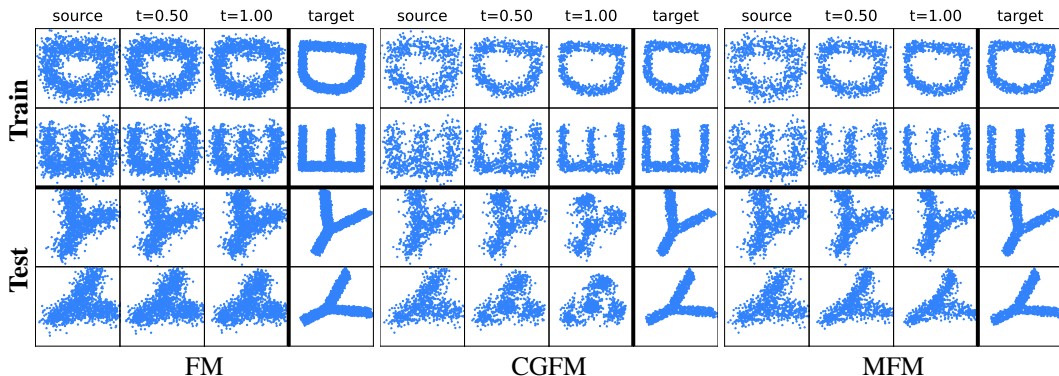

Figure 2: Examples of model-generated samples for synthetic letters from the source distribution ($t = 0$) to predicted target distribution ($t = 1$). See Fig. 4 in Appendix F for a larger set of examples.

Table 1: Results of the synthetic letters experiment for population prediction on seen train populations and unseen test populations. We report the the 1-Wasserstein ($\mathcal{W}_1$), 2-Wasserstein ($\mathcal{W}_2$), and the maximum-mean-discrepancy (MMD) distributional distances. We consider 4 settings for MFM with varying $k$.

| | **Train** | | | **Test** | | |
|---|---|---|---|---|---|---|
| | $\mathcal{W}_1$ | $\mathcal{W}_2$ | MMD ($\times 10^{-3}$) | $\mathcal{W}_1$ | $\mathcal{W}_2$ | MMD ($\times 10^{-3}$) |
| FM | $0.216 \pm 0.000$ | $0.280 \pm 0.000$ | $2.38 \pm 0.00$ | $0.237 \pm 0.000$ | $0.315 \pm 0.000$ | $\mathbf{3.28 \pm 0.00}$ |
| CGFM | $\mathbf{0.093 \pm 0.000}$ | $\mathbf{0.112 \pm 0.000}$ | $0.34 \pm 0.00$ | $0.317 \pm 0.000$ | $0.397 \pm 0.000$ | $6.67 \pm 0.00$ |
| MFM ($k = 0$) | $0.099 \pm 0.000$ | $0.128 \pm 0.000$ | $0.25 \pm 0.00$ | $0.221 \pm 0.000$ | $0.267 \pm 0.000$ | $3.77 \pm 0.00$ |
| MFM ($k = 1$) | $\mathbf{0.096 \pm 0.003}$ | $0.124 \pm 0.004$ | $\mathbf{0.22 \pm 0.04}$ | $0.217 \pm 0.003$ | $0.261 \pm 0.003$ | $3.80 \pm 0.28$ |
| MFM ($k = 10$) | $\mathbf{0.096 \pm 0.003}$ | $0.124 \pm 0.003$ | $\mathbf{0.23 \pm 0.04}$ | $\mathbf{0.213 \pm 0.008}$ | $\mathbf{0.256 \pm 0.008}$ | $\mathbf{3.68 \pm 0.45}$ |
| MFM ($k = 50$) | $0.099 \pm 0.003$ | $0.127 \pm 0.003$ | $0.25 \pm 0.05$ | $0.226 \pm 0.005$ | $0.270 \pm 0.007$ | $4.38 \pm 0.30$ |

(GCN) with a $k$-nearest neighbour graph edge pooling layer. We include details regarding model hyperparameters, training/optimization, and implementation in Appendix B and Appendix B.2. The results for all the models are averaged over three random seeds.

## 5.1 Synthetic Experiment

We curate a synthetic dataset of the joint distributions $\{(p_0(x_0, | i), p_1(x_1 | i))\}_{i=1}^N$ by simulating a diffusion process applied to a set of pre-defined target distributions $p_1(x_1 | i)$ for $i = 1, \ldots, N$. To get a paired population $p_0(x_0 | i)$ we simulate the forward diffusion process without drift $x_0 \sim \mathcal{N}(x_1, \sigma)$. After this setup, for reasonable values of $\sigma$, we assume that one can reverse the diffusion process and learn the push-forward map from $p_0(x_0 | i)$ to $p_1(x_1 | i)$ for every index $i$. For this task, given the $i$-th population index we denote $p_0(x_0 | i)$ as the *source* population $p_1(x_1 | i)$ as the $i$-th *target* population.

To construct $p_1(x_1 | i)$, we discretize samples from a defined silhouette; e.g. an image of a character, where $i$ indexes the respective character. We use upper case letters as the silhouette and generate the corresponding samples $x_1 \sim p_1(x_1 | i)$ from the uniform distribution over the silhouette and run the diffusion process for samples $x_1$ to acquire $x_0$. We construct the *training data* using 10 random orientations of 24 letters, while only using the upright orientation for the remaining letters "X" and "Y". We construct the *test data* by using 10 random orientations of "X" and "Y" (validation and test, respectively) that differ from the upright orientations of the same letters in the training data. We do this to simplify the generalization task – the model will see the shapes of "X" and "Y" during training, but the same letters under different orientations remain unseen.

We train FM, CGFM and 4 variants of MFM of varying $k$ for the GCN population embedding model $\varphi(p_t; \theta, k)$. When $k = 0$, $\varphi(p_t; \theta, k)$ becomes identical to the DeepSets model (Zaheer et al., 2017). We compare MFM to Flow-Matching (FM) and Conditional Generation via Flow-Matching (CGFM). FM does not have access to conditional information; hence will only learn an aggregated lens of the distribution dynamics and will not be able to fit the training data, and consequently won't generalize to the test conditions. For the training data, CGFM vector field model takes in the distribution index $i$ as a one-hot input condition. On the test set, since none of these indices is present, we input the normalized constant vector, which averages the learned embeddings of the indices. Because of this, CGFM will fit the training data, however, will not be able to generalize to the unseen condition in the test dataset. Note that the CGFM can be viewed as an *idealized* model for the train data since

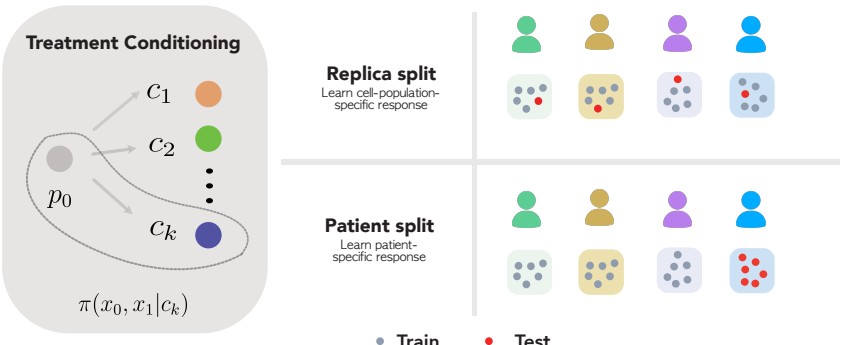

Figure 3: Organoid drug-screen dataset overview. *Left*: a given replica consists of a control distribution $p_0$ and corresponding treatment response distribution $p_1$ for treatment condition $c_i$. *Right*: train and test data splits for replica (top) and patients (bottom) splits, restively. For each experiment there are 11 treatments, 10 patients and 3 culture conditions.

it gets perfect information regarding the population conditions. We use CGFM to assess if other models are fitting the data. For MFM, we expect to both fit the training data and generalize to unseen distributional conditions.

In Fig. 2, we observe that indeed FM fails to adequately learn to sample from $p_1(x_1 \mid i)$ in the training set, and likewise fails to generalize, while CGFM is able to effectively sample from $p_1(x_1 \mid i)$ in the training set, but fails to generalize. We report results for the synthetic experiment in Table 1. As expected, CGFM fits the training data, however, fails to generalize beyond its set of training conditions. In contrast, we see that MFM is able to both fit the training data (approaching the performance of CGFM) while also generalizing to the unseen test distributions. FM fails to fit the train data and fails to generalize under the test conditions. Interestingly, although MFM performs better for certain values of $k$ versus others, overall performance does not vary significantly for the range considered.

## 5.2 Experiments on Organoid Drug-screen Data

**Data.** For experiments on biological data, we use the organoid drug-screen dataset from Ramos Zapatero et al. (2023). This dataset is a single-cell mass-cytometry dataset collected over 10 patients. Somewhat unique to this dataset, unlike many prior perturbation-screen datasets which have a single control population, this dataset has matched controls to each experimental condition. Populations from each patient are treated with 11 different drug treatments of varying dose concentrations.[1] We use the term *replicate* to define control-treatment population pairs, $p_0(x_0 \mid c_i)$ and $p_1(x_1 \mid c_i)$, respectively (see Fig. 3-left). In each patient, cell population are categorized into 3 cell *cultures*: (i) cancer associated Fibroblasts, (ii) patient-derived organoid cancer cells (PDO), and (iii) patient-derived organoid cancer cells co-cultured fibroblasts (PDOF). We report results averaged over Fibroblast/PDO/PDOF cultures and results for the individual cultures (this is reported in Appendix F).

**Pre-processing and data splits.** We filter each cell population to contain at least 1000 cells and consider 43 bio-markers. We consider two data splits for the organoid drug-screen dataset (see Fig. 3-right). (1) *Replicate split*; here we leave-out replicates evenly across all patients for testing. (2) *Patients split*; here we leave-out replicates fully in one patients – in this setting, we are testing the ability of of model to generalize population prediction of treatment response for unseen patients. In both settings, we normalize the data and embed it into a lower dimensional principle components (PC) representation. We do this to reduce the dimensionality of the data and to extract the relevant information from the 43 bio-markers (features) of the ambient space. We train and evaluate all models in the PC space. For all organoid drug-screen dataset experiments we use PC=10. Further details regarding data pre-processing and data splits are provided in Appendix B.2.

For the organoid drug-screen experiments, we consider an ICNN architecture in addition to the Flow-matching models. The ICNN model is based on CellOT (Bunne et al., 2023a); a method for learning cell specific response to treatments. The ICNN (and likewise CellOT) counterparts our FM

---

[1]We consider only the highest dosage and leave exploration of dose-dependent response to future work.

Table 2: Experimental results on the organoid drug-screen dataset for population prediction of treatment response across *replicate* populations averaged over co-culture conditions. Results are reported for models trained on data embedded into 10 principle components. We report the the 1-Wasserstein ($\mathcal{W}_1$), 2-Wasserstein ($\mathcal{W}_2$), and the maximum-mean-discrepancy (MMD) distributional distances. We consider two settings for MFM with varying nearest-neighbours parameter. For extended results in Table 4.

| | Train | | | Test | | |
|---|---|---|---|---|---|---|
| | $\mathcal{W}_1$ | $\mathcal{W}_2$ | MMD ($\times 10^{-3}$) | $\mathcal{W}_1$ | $\mathcal{W}_2$ | MMD ($\times 10^{-3}$) |
| FM | $1.946 \pm 0.083$ | $2.178 \pm 0.092$ | $6.32 \pm 0.36$ | $2.087 \pm 0.035$ | $2.301 \pm 0.043$ | $9.29 \pm 0.77$ |
| ICNN | $2.112 \pm 0.012$ | $2.317 \pm 0.011$ | $190.17 \pm 4.87$ | $2.200 \pm 0.011$ | $2.395 \pm 0.010$ | $249.33 \pm 4.67$ |
| CGFM | $\mathbf{1.823 \pm 0.126}$ | $\mathbf{2.009 \pm 0.143}$ | $\mathbf{4.16 \pm 1.00}$ | $2.213 \pm 0.137$ | $2.416 \pm 0.154$ | $13.91 \pm 2.41$ |
| MFM ($k = 0$) | $1.829 \pm 0.050$ | $2.012 \pm 0.058$ | $4.64 \pm 0.66$ | $1.959 \pm 0.050$ | $2.144 \pm 0.059$ | $7.35 \pm 1.20$ |
| MFM ($k = 10$) | $1.842 \pm 0.049$ | $2.020 \pm 0.057$ | $4.76 \pm 0.66$ | $\mathbf{1.954 \pm 0.047}$ | $\mathbf{2.136 \pm 0.052}$ | $\mathbf{7.34 \pm 0.93}$ |

Table 3: Experimental results on the organoid drug-screen dataset for population prediction of treatment response across *patient* populations. Results shown in this table are broken out in Table 5.

| | Train | | | Test | | |
|---|---|---|---|---|---|---|
| | $\mathcal{W}_1$ | $\mathcal{W}_2$ | MMD ($\times 10^{-3}$) | $\mathcal{W}_1$ | $\mathcal{W}_2$ | MMD ($\times 10^{-3}$) |
| FM | $1.995 \pm 0.138$ | $2.246 \pm 0.193$ | $6.87 \pm 2.65$ | $2.607 \pm 0.028$ | $2.947 \pm 0.050$ | $21.58 \pm 1.02$ |
| ICNN | $2.163 \pm 0.067$ | $2.367 \pm 0.070$ | $192.67 \pm 4.22$ | $2.702 \pm 0.027$ | $2.996 \pm 0.033$ | $452.67 \pm 19.14$ |
| CGFM | $\mathbf{1.773 \pm 0.072}$ | $\mathbf{1.954 \pm 0.092}$ | $\mathbf{3.03 \pm 0.69}$ | $2.675 \pm 0.019$ | $2.938 \pm 0.020$ | $23.75 \pm 0.61$ |
| MFM ($k = 0$) | $1.863 \pm 0.056$ | $2.048 \pm 0.063$ | $5.01 \pm 0.53$ | $2.393 \pm 0.160$ | $2.685 \pm 0.122$ | $16.66 \pm 1.99$ |
| MFM ($k = 10$) | $1.881 \pm 0.071$ | $2.074 \pm 0.091$ | $5.25 \pm 0.78$ | $\mathbf{2.326 \pm 0.072}$ | $\mathbf{2.610 \pm 0.073}$ | $\mathbf{14.30 \pm 2.27}$ |

model in that it does not take the population index $i$ as a condition. Therefore, it will neither be able to fit the training data, nor generalize.

**Predicting treatment response across replicates.** We show results for generalization across replicates in Table 2. As expected, we observe that CGFM fits the training data, but does not generalize to the test replicates. With this, we can observe that the FM and ICNN models fail to fit the train data, relative to CGFM, and also fail to generalize. MFM ($k = 10$) performs best on generalization to unseen replicates. We include results reported for the separate cell cultures in Table 4 in Appendix F.

**Predicting treatment response across patients.** We show results for generalization across patients in Table 3. Similar to the replicates data setting, we observe that CGFM fits the training data, but does not generalize to the test replicates. Likewise, the FM and ICNN models fail to fit the train data, relative to CGFM, and also fail to generalize. MFM ($k = 10$) performs best on generalization to unseen replicates. We include results reported for the separate cell cultures in Table 5 in Appendix F.

Through the biological and synthetic experiments, we have shown that MFM is able to generalize to unseen distributions/populations. The implication of our results suggest that MFM can learn population dynamics in unseen environments. In biological contexts, like the one we have shown in this work, this result indicates that we can learn population dynamics, of treatment response or any arbitrary perturbation, in new/unseen patients. This works towards a model where it is possible to predict and design an individualized treatment regimen for each patient based on their individual characteristics and tumor microenvironment.

# 6 Conclusion and Future Work

Our paper highlights the significance of modeling dynamics based on the entire distribution. While flow-based models offer a promising avenue for learning dynamics at the population level, they were previously restricted to a single initial population and predefined conditions.

In this paper, we introduce Meta Flow Matching (MFM) as a practical solution to address these limitations. By integrating along vector fields of the Wasserstein manifold, MFM allows for a more comprehensive model of dynamical systems with interacting particles. Crucially, MFM leverages graph neural networks to embed the initial population, enabling the model to generalize over various initial distributions. MFM opens up new possibilities for understanding complex phenomena that emerge from interacting systems in biological and physical systems.

In practice, we demonstrate that MFM learns meaningful embeddings of single-cell populations along with the developmental model of these populations. Moreover, our empirical study demonstrates the possibility of modeling patient-specific response to treatments via the meta-learning.

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

## A  Proof of Theorem 1

**Theorem 1.** *Consider a dataset of populations $\mathcal{D} = \{(\pi(x_0, x_1 \mid i))\}_i$ generated from some unknown conditional model $\pi(x_0, x_1 \mid c)p(c)$. Then the following objective*

$$\mathcal{L}(\omega, \theta) = \ \mathbb{E}_{p(c)} \int_0^1 dt \ \mathbb{E}_{p_t(x_t \mid c)} \| v_t^*(x_t \mid c) - v_t(x_t \mid \varphi(p_0, \theta), \omega) \|^2 \tag{16}$$

*is equivalent to the Meta Flow Matching objective*

$$\mathcal{L}_{MFM}(\omega, \theta) = \ \mathbb{E}_{i \sim \mathcal{D}} \mathbb{E}_{\pi(x_0, x_1 \mid i)} \int_0^1 dt \ \left\| \frac{\partial}{\partial t} f_t(x_0, x_1) - v_t(f_t(x_0, x_1) \mid \varphi(p_0, \theta); \omega) \right\|^2 \tag{17}$$

*up to an additive constant.*

*Proof.* The loss function

$$\mathcal{L}(\omega, \theta) = \ \mathbb{E}_{p(c)} \int_0^1 dt \ \mathbb{E}_{p_t(x_t \mid c)} \| v_t^*(x_t \mid c) - v_t(x_t \mid \varphi(p_t, \theta); \omega) \|^2 \tag{20}$$

$$= \ - 2\mathbb{E}_{p(c)} \int dt dx \ \langle p_t(x \mid c) v_t^*(x \mid c), v_t(x \mid \varphi(p_t, \theta); \omega) \rangle + \tag{21}$$

$$+ \ \mathbb{E}_{p(c)} \int_0^1 dt \ \mathbb{E}_{p_t(x_t \mid c)} \| v_t(x_t \mid \varphi(p_t, \theta), \omega) \|^2 + \tag{22}$$

$$+ \ \mathbb{E}_{p(c)} \int_0^1 dt \ \mathbb{E}_{p_t(x_t \mid c)} \| v_t^*(x_t \mid c) \|^2 \, . \tag{23}$$

The last term does not depend on $\theta$, the second term we can estimate, for the first term, we use the formula for the (from Eq. (8))

$$p_t(\xi \mid c) v_t^*(\xi \mid c) = \mathbb{E}_{\pi(x_0, x_1)} \delta(f_t(x_0, x_1) - \xi) \frac{\partial f_t(x_0, x_1)}{\partial t} \, . \tag{24}$$

Thus, the loss is equivalent (up to a constant) to

$$\mathcal{L}(\omega, \theta) = \ - 2\mathbb{E}_{p(c)} \mathbb{E}_{\pi(x_0, x_1 \mid c)} \int dt \ \left\langle \frac{\partial f_t(x_0, x_1)}{\partial t}, v_t(f_t(x_0, x_1) \mid \varphi(p_t, \theta); \omega) \right\rangle + \tag{25}$$

$$+ \ \mathbb{E}_{p(c)} \mathbb{E}_{\pi(x_0, x_1 \mid c)} \int_0^1 dt \ \| v_t(f_t(x_0, x_1) \mid \varphi(p_t, \theta), \omega) \|^2 \pm \tag{26}$$

$$\pm \ \mathbb{E}_{p(c)} \mathbb{E}_{\pi(x_0, x_1 \mid c)} \int_0^1 dt \ \left\| \frac{\partial f_t(x_0, x_1)}{\partial t} \right\|^2 \tag{27}$$

$$= \ \mathbb{E}_{c \sim p(c)} \mathbb{E}_{\pi(x_0, x_1 \mid c)} \int_0^1 dt \ \left\| \frac{\partial}{\partial t} f_t(x_0, x_1) - v_t(f_t(x_0, x_1) \mid \varphi(p_t, \theta); \omega) \right\|^2 \, . \tag{28}$$

Note that in the final expression we do not need access to the probabilistic model of $p(c)$ if the joints $\pi(x_0, x_1 \mid c)$ are already sampled in the data $\mathcal{D}$. Thus, we have

$$\mathcal{L}(\omega, \theta) = \ \mathbb{E}_{c \sim p(c)} \mathbb{E}_{\pi(x_0, x_1 \mid c)} \int_0^1 dt \ \left\| \frac{\partial}{\partial t} f_t(x_0, x_1) - v_t(f_t(x_0, x_1) \mid \varphi(p_t, \theta); \omega) \right\|^2 \tag{29}$$

$$= \ \mathbb{E}_{i \sim \mathcal{D}} \mathbb{E}_{\pi(x_0, x_1 \mid i)} \int_0^1 dt \ \left\| \frac{\partial}{\partial t} f_t(x_0, x_1) - v_t(f_t(x_0, x_1) \mid \varphi(p_t, \theta); \omega) \right\|^2 \tag{30}$$

$$= \ \mathcal{L}_{MFM}(\omega, \theta) \, . \tag{31}$$

$\square$

## B  Experimental Details

### B.1  Synthetic letters data

The synthetic letters dataset contains 242 train populations a 10 test populations. Each population contains roughly between 750 and 2700 samples. In this dataset.

### B.2 Organoid drug-screen data

The organoid drug-screen dataset contains a total of 927 replicates (or coupled populations). In the *replicates split*, we use 713 populations for training and 103 left-out populations for testing. In the *patients split*, we use 861 populations for training and 33 left-out populations for testing.

### B.3 Model architectures and hyperparameters

**ICNN.** The ICNN baseline was constructed with two networks ICNN network $f(x)$ and $g(x)$, with non-negative leaky ReLU activation layers. $f(x)$ is used to minimize the transport distance and $g(x)$ is used to transport from source to target. It has four hidden units with width of 64, and a latent dimension of 50. Both networks uses Adam optimizer (lr=$1e-4$, $\beta_1$=0.5, $\beta_2$=0.9). $g(x)$ is trained with an inner iteration of 10 for every iteration $f(x)$ is trained.

**Vector Field Models.** All vector field models $v_t$ are parameterized 4 linear layers with 512 hidden units and SELU activation functions. The FM vector field model additionally takes a conditional input for the one-hot treatment encoding. CGFM takes the conditional input for the one-hot treatment conditions as well as a one-hot encoding for the population index condition $i$. The MFM vector field model takes population embedding conditions, that is output from the GCN, as input, as well as the treatment one-hot encoding. All vector field models use temporal embeddings for time and positional embeddings for the input samples. We did not sweep the size of this embeddings space and found that a temporal embedding and positional embeddings sizes of 128 worked sufficiently well.

**Graph Neural Network.** We considered a GCN model that consists of a $k$-nearest neighbour graph edge pooling layer and 3 graph convolution layers with 512 hidden units. The final GCN model layer outputs an embedding representation $e \in \mathbb{R}^d$. For the Synthetic experiment, we found that $d = 256$ performed well, and $d = 128$ performed well for the biological experiments. We normalize and project embeddings onto a hyper-sphere, and find that this normalization helps improve training. Additionally, the GCN takes a one-hot cell-type encoding (encoding for Fibroblast cells or PDO cells) for the control populations $p_0$. This may be beneficial for PDOF populations where both Fibroblast cells and PDO cells are present. However, it is important to note that labeling which cells are Fibroblasts versus PDOs withing the PDOF cultures is difficult and noisy in itself, hence such a cell-type condition may yield no additive information/performance gain.

**Optimization.** We use the Adam optimizer with a learning rate of $0.0001$ for all Flow-matching models (FM, CGFM, MFM). We also used the Adam optimizer with a learning rate of $0.0001$ for the GCN model. To train the MFM (FM+GCN) models, we alternate between updating the vector field model parameters $\omega$ and the GCN model parameters $\theta$. We alternate between updating the respective model parameters every epoch. FM and CGFM model were trained for 2000 epochs, while MFM models were trained for 4000 epochs. Due to the alternating optimization, the MFM vector field model receives half as many updates compared to its counterparts (FM and CGFM). Therefore, training for the double the epochs is necessary for fair comparison.

The hyperparameters stated in this section were selected from brief and small grid search sweeps. We did not conduct any thorough hyperparameter optimization.

## C Implementation Details

We implement all our experiments using PyTorch and PyTorch Geometric. We submitted our code as supplementary material with our submission.

All experiments were conducted on a HPC cluster primarily on NVIDIA Tesla T4 16GB GPUs. Each individual seed experiment run required only 1 GPU. Each experiment ran between 3-11 hours and all experiments took approximately 500 GPU hours.

## D Limitations

In this work we explored empirically the effect of conditioning the learned flow on the initial distribution. We argue this is a more natural model for many biological systems. However, there are many other aspects of modeling biological systems that we did not consider. In particular we

did not consider extensions to the manifold setting (Huguet et al., 2022, 2023), unbalanced optimal transport (Benamou, 2003; Yang and Uhler, 2019; Chizat et al., 2018), aligned (Somnath et al., 2023; Liu et al., 2023), or stochastic settings (Bunne et al., 2023a; Koshizuka and Sato, 2023) in this work.

# E    Broader Impacts

This paper is primarily a theoretical and methodological contribution with little societal impact. MFM can be used to better model dynamical systems of interacting particles and in particular cellular systems. Better modeling of cellular systems can potentially be used for the development of malicious biological agents. However, we do not see this as a significant risk at this time.

# F    Extended Results

Table 4: Experimental results on the organoid drug-screen dataset for population prediction of treatment response across **replicate** populations. Results are reported for models trained on data embedded into 10 principle components. We report the the 1-Wasserstein ($\mathcal{W}_1$), 2-Wasserstein ($\mathcal{W}_2$), and the maximum-mean-discrepancy (MMD) distributional distances. We consider 2 settings for MFM with varying nearest-neighbours parameter.

| | | **Fibroblasts** | | | | |
| | | Train | | | Test | |
| | $\mathcal{W}_1$ | $\mathcal{W}_2$ | MMD ($\times 10^{-3}$) | $\mathcal{W}_1$ | $\mathcal{W}_2$ | MMD ($\times 10^{-3}$) |
|---|---|---|---|---|---|---|
| FM | $1.584 \pm 0.022$ | $1.730 \pm 0.015$ | $3.12 \pm 0.59$ | $1.612 \pm 0.014$ | $1.736 \pm 0.024$ | $3.62 \pm 0.15$ |
| ICNN | $1.613 \pm 0.010$ | $1.703 \pm 0.010$ | $52.4 \pm 1.64$ | $1.655 \pm 0.008$ | $1.746 \pm 0.008$ | $53.0 \pm 5.00$ |
| CGFM | $\mathbf{1.472 \pm 0.046}$ | $\mathbf{1.548 \pm 0.048}$ | $\mathbf{1.28 \pm 0.74}$ | $1.633 \pm 0.022$ | $1.724 \pm 0.023$ | $4.95 \pm 0.72$ |
| MFM ($k = 0$) | $1.519 \pm 0.034$ | $1.599 \pm 0.036$ | $2.56 \pm 0.56$ | $\mathbf{1.574 \pm 0.002}$ | $\mathbf{1.657 \pm 0.003}$ | $\mathbf{3.31 \pm 0.12}$ |
| MFM ($k = 10$) | $1.547 \pm 0.027$ | $1.617 \pm 0.027$ | $2.84 \pm 0.56$ | $1.576 \pm 0.017$ | $1.658 \pm 0.019$ | $3.44 \pm 0.19$ |
| | | **PDO** | | | | |
| | | Train | | | Test | |
| | $\mathcal{W}_1$ | $\mathcal{W}_2$ | MMD ($\times 10^{-3}$) | $\mathcal{W}_1$ | $\mathcal{W}_2$ | MMD ($\times 10^{-3}$) |
| FM | $2.002 \pm 0.027$ | $2.201 \pm 0.025$ | $6.40 \pm 0.10$ | $2.033 \pm 0.015$ | $2.210 \pm 0.016$ | $6.92 \pm 0.65$ |
| ICNN | $2.29 \pm 0.005$ | $2.458 \pm 0.003$ | $245.8 \pm 9.18$ | $2.247 \pm 0.005$ | $2.415 \pm 0.004$ | $153 \pm 1.00$ |
| CGFM | $1.818 \pm 0.198$ | $1.931 \pm 0.229$ | $3.78 \pm 0.27$ | $2.255 \pm 0.216$ | $2.434 \pm 0.240$ | $12.16 \pm 3.87$ |
| MFM ($k = 0$) | $1.817 \pm 0.043$ | $1.935 \pm 0.040$ | $\mathbf{3.61 \pm 0.50}$ | $1.909 \pm 0.076$ | $2.057 \pm 0.098$ | $\mathbf{5.14 \pm 0.92}$ |
| MFM ($k = 10$) | $\mathbf{1.805 \pm 0.074}$ | $\mathbf{1.921 \pm 0.078}$ | $3.68 \pm 0.78$ | $\mathbf{1.903 \pm 0.068}$ | $\mathbf{2.051 \pm 0.084}$ | $\mathbf{5.14 \pm 0.90}$ |
| | | **PDOF** | | | | |
| | | Train | | | Test | |
| | $\mathcal{W}_1$ | $\mathcal{W}_2$ | MMD ($\times 10^{-3}$) | $\mathcal{W}_1$ | $\mathcal{W}_2$ | MMD ($\times 10^{-3}$) |
| FM | $2.252 \pm 0.20$ | $2.603 \pm 0.236$ | $9.43 \pm 0.38$ | $2.616 \pm 0.076$ | $2.958 \pm 0.089$ | $19.34 \pm 1.51$ |
| ICNN | $2.432 \pm 0.021$ | $2.791 \pm 0.020$ | $272.3 \pm 3.80$ | $2.699 \pm 0.021$ | $3.023 \pm 0.019$ | $542 \pm 8.00$ |
| CGFM | $2.179 \pm 0.133$ | $2.548 \pm 0.153$ | $\mathbf{7.42 \pm 2.00}$ | $2.750 \pm 0.173$ | $3.089 \pm 0.200$ | $22.63 \pm 2.64$ |
| MFM ($k = 0$) | $\mathbf{2.150 \pm 0.073}$ | $\mathbf{2.502 \pm 0.099}$ | $7.75 \pm 0.93$ | $2.395 \pm 0.071$ | $2.717 \pm 0.076$ | $13.61 \pm 2.56$ |
| MFM ($k = 10$) | $2.174 \pm 0.046$ | $2.523 \pm 0.067$ | $7.75 \pm 0.65$ | $\mathbf{2.382 \pm 0.055}$ | $\mathbf{2.699 \pm 0.054}$ | $\mathbf{13.45 \pm 1.69}$ |

Table 5: Experimental results on the organoid drug-screen dataset for population prediction of treatment response across **patient** populations. Results are reported for models trained on data embedded into 10 principle components. We report the the 1-Wasserstein ($\mathcal{W}_1$), 2-Wasserstein ($\mathcal{W}_2$), and the maximum-mean-discrepancy (MMD) distributional distances. We consider 2 settings for MFM with varying nearest-neighbours parameter.

| | **Fibroblasts** | | | | | |
| | | Train | | | Test | |
| | $\mathcal{W}_1$ | $\mathcal{W}_2$ | MMD ($\times 10^{-3}$) | $\mathcal{W}_1$ | $\mathcal{W}_2$ | MMD ($\times 10^{-3}$) |
|---|---|---|---|---|---|---|
| FM | $1.599 \pm 0.071$ | $1.761 \pm 0.137$ | $2.82 \pm 0.34$ | $1.667 \pm 0.003$ | $1.846 \pm 0.064$ | $7.85 \pm 0.15$ |
| ICNN | $1.695 \pm 0.08$ | $1.796 \pm 0.09$ | $48.2 \pm 3.412$ | $1.6 \pm 0.009$ | $1.68 \pm 0.013$ | $62.2 \pm 1.32$ |
| CGFM | $\mathbf{1.496 \pm 0.019}$ | $\mathbf{1.572 \pm 0.016}$ | $\mathbf{1.45 \pm 0.14}$ | $1.566 \pm 0.028$ | $1.652 \pm 0.026$ | $6.46 \pm 0.82$ |
| MFM ($k=0$) | $1.551 \pm 0.037$ | $1.632 \pm 0.042$ | $2.31 \pm 0.71$ | $1.453 \pm 0.200$ | $1.527 \pm 0.022$ | $3.66 \pm 0.67$ |
| MFM ($k=10$) | $1.555 \pm 0.034$ | $1.635 \pm 0.039$ | $2.54 \pm 0.42$ | $\mathbf{1.441 \pm 0.003}$ | $\mathbf{1.514 \pm 0.001}$ | $\mathbf{3.37 \pm 0.72}$ |
| | **PDO** | | | | | |
| | | Train | | | Test | |
| | $\mathcal{W}_1$ | $\mathcal{W}_2$ | MMD ($\times 10^{-3}$) | $\mathcal{W}_1$ | $\mathcal{W}_2$ | MMD ($\times 10^{-3}$) |
| FM | $1.996 \pm 0.196$ | $2.171 \pm 0.243$ | $6.79 \pm 3.40$ | $2.128 \pm 0.064$ | $2.312 \pm 0.075$ | $7.88 \pm 1.26$ |
| ICNN | $2.315 \pm 0.060$ | $2.478 \pm 0.057$ | $236.8 \pm 0.006$ | $2.538 \pm 0.018$ | $2.731 \pm 0.027$ | $232.8 \pm 20.6$ |
| CGFM | $\mathbf{1.662 \pm 0.026}$ | $\mathbf{1.760 \pm 0.023}$ | $\mathbf{1.74 \pm 0.16}$ | $2.460 \pm 0.018$ | $2.533 \pm 0.023$ | $13.6 \pm 0.25$ |
| MFM ($k=0$) | $1.837 \pm 0.058$ | $1.964 \pm 0.059$ | $3.74 \pm 0.29$ | $2.010 \pm 0.142$ | $2.168 \pm 0.182$ | $6.01 \pm 1.77$ |
| MFM ($k=10$) | $1.838 \pm 0.035$ | $1.957 \pm 0.038$ | $3.75 \pm 0.41$ | $\mathbf{1.971 \pm 0.082}$ | $\mathbf{2.114 \pm 0.101}$ | $\mathbf{5.42 \pm 1.11}$ |
| | **PDOF** | | | | | |
| | | Train | | | Test | |
| | $\mathcal{W}_1$ | $\mathcal{W}_2$ | MMD ($\times 10^{-3}$) | $\mathcal{W}_1$ | $\mathcal{W}_2$ | MMD ($\times 10^{-3}$) |
| FM | $2.390 \pm 0.148$ | $2.806 \pm 0.198$ | $11.0 \pm 2.21$ | $4.026 \pm 0.018$ | $4.683 \pm 0.011$ | $49.0 \pm 1.66$ |
| ICNN | $2.479 \pm 0.06$ | $2.826 \pm 0.063$ | $291 \pm 9.24$ | $3.968 \pm 0.0554$ | $4.579 \pm 0.060$ | $1263 \pm 37.5$ |
| CGFM | $\mathbf{2.160 \pm 0.170}$ | $\mathbf{2.530 \pm 0.237}$ | $\mathbf{7.90 \pm 1.79}$ | $4.000 \pm 0.010$ | $4.629 \pm 0.012$ | $49.2 \pm 0.76$ |
| MFM ($k=0$) | $2.202 \pm 0.072$ | $2.548 \pm 0.089$ | $8.98 \pm 0.59$ | $3.717 \pm 0.138$ | $4.360 \pm 0.162$ | $40.3 \pm 3.52$ |
| MFM ($k=10$) | $2.251 \pm 0.143$ | $2.631 \pm 0.197$ | $9.45 \pm 1.52$ | $\mathbf{3.565 \pm 0.132}$ | $\mathbf{4.201 \pm 0.119}$ | $\mathbf{36.1 \pm 4.97}$ |

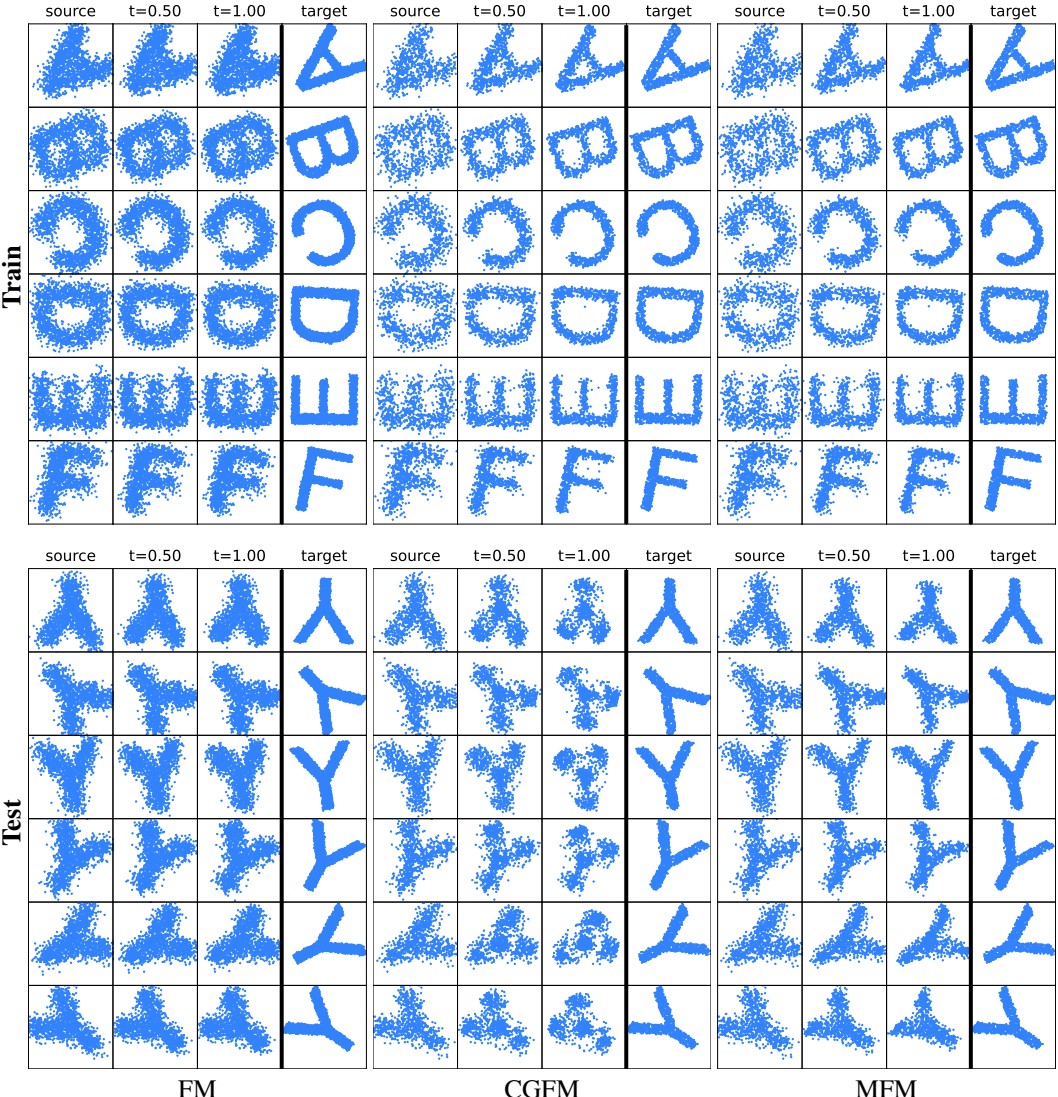

Figure 4: Model-generated samples for synthetic letters from the source ($t = 0$) to target ($t = 1$) distributions.

