# OpenReview forum: "Meta Flow Matching: Integrating Vector Fields on the Wasserstein Manifold"
_NeurIPS.cc/2024/Conference — Submitted to NeurIPS 2024_

### Official Review · Reviewer_BiEg · 2024-07-10

**Soundness:** 4
**Presentation:** 4
**Contribution:** 3
**Rating:** 7
**Confidence:** 4

**Summary:**

This paper addresses the issue of the flow matching method's lack of dependence on the data population. It proposes incorporating the initial population density into the vector field through amortization—using a Graph Neural Network (GNN) to embed the populations and adding this embedding to the input of the vector field network. The paper argues that this dependence would better model the data due to sample interactions, demonstrating improved generalization on unseen initial distributions. The method's application is showcased in perturbation drug screening.

**Strengths:**

### Originality
The problem setting of adding population dependence to flow matching is novel. The model framework of adding input to the vector field network using a GNN as a population encoder is also novel.

### Clarity

The paper is clearly written, with rigorous mathematical notations. The related work and introductions are especially well-written.

### Quality

The writing is good, and the experiment involves many baselines.

### Significance

The proposed method excels at generalizing to unseen populations, which is a significant improvement over existing methods, particularly when the conditions for generation are complex. The application on drug screening addresses a significant scientific problem and holds promise for personalized healthcare.

**Weaknesses:**

1. The paper could explain more about the meta-learning aspect of this method.
2. It could include explanations and/or ablation studies on the role of meta-learning and the GNN, especially in the synthetic experiment.
3. More detail is needed on what properties of the Wasserstein manifold of probabilities are used in the model. It is unclear how the model proposed in section 3.2 depends on the properties of the Wasserstein manifold described in section 3.1.

**Questions:**

1. How is this model different from using a GNN encoder to add the embedding to the input of the vector field, trained end-to-end? Specifically, what makes it meta-learning? Considering that $\theta$ and $\omega$ are jointly optimized to minimize $\mathcal{L}_{MFM}$, is there anything unique about the training algorithm?
2. How would this meta-learning framework compare with using the embedding from a pretrained encoder, such as a VGAE, or a classification model?
3. For the synthetic experiment, how much does the orientation-invariant embedding from the GNN help the model? Is the observed "generalization" due to the unique inductive bias of the GNN? Would any other encoder, without orientation invariance, such as an MLP, still yield good performance?
4. How well does it scale to high dimensions, considering that only 10 dimensions are used in the experiments? How well does it scale to large datasets where you need to build large KNN graphs? Would it scale effectively to high-dimensional tasks such as image generation?
5. In line 261, it states, “We use CGFM to assess if other models are fitting the data.” Is this used as an evaluation metric in the experiments?

**Limitations:**

The authors have adequately addressed the limitations.

---

> ### Author Rebuttal · Authors · 2024-08-07
>
> >The paper could explain more about the meta-learning aspect of this method. Specifically, what makes it meta-learning?
>
> Our work follows the naming convention of *Meta Optimal Transport* [1] and is "meta" in the sense of amortizing learning over multiple input distributions. This is related to how "meta" is used in the meta learning setting — meta learning solves multiple learning problems simultaneously while meta optimal transport and meta flow matching solves multiple OT or flow matching problems simultaneously. We added a section on other "meta"-based approaches in our related works and clarified our notion of "meta" in the text.
>
> >It could include explanations and/or ablation studies on the role of meta-learning and the GNN, especially in the synthetic experiment.
>
> We have included a primary ablation study in the main experiments (on synthetic and real data) over the KNN parameter $k$ (see the general response). We are happy to include further ablations.
>
> >detail is needed on what properties of the Wasserstein manifold of probabilities are used in the model.
>
> In lines 150-152, we describe our working assumptions which motivate the usage of the Wasserstein manifold geometry. We assume that (i) we solve a regression problem on the space of marginals. A single point of space is a distribution (ii) the ground true evolutions of the marginals happen according to the continuity equation, i.e. the tangent space corresponds to the one of the Wasserstein manifold and there is no birth\death processes (iii) the final marginal can be uniquely defined from the initial marginal, which corresponds to the existence of an ODE on the Wasserstein manifold with non-intersecting integral curves.
>
> >How is this model different from using a GNN encoder to add the embedding to the vector field, trained end-to-end?
>
> We train the flow model $v_t(\cdot;\omega)$ and the embedding model $\varphi(\cdot;\theta)$ jointly using the flow matching loss (see supplementary PDF Algorithm 1). We can efficiently update model parameters $\omega$ and $\theta$ through alternating gradient updates while using the same loss. This is a desirable property since we don't need to pre-train the GNN encoder or decide on a different loss. We leave further investigation on the distribution embedding model for future work.
>
> >Considering that $\theta$ and $\omega$ are jointly optimized to minimize $\mathcal{L}_{MFM}$, is there anything unique about the training algorithm?
>
> This training algorithm is unique because we use the flow matching loss to train both the flow model (which is a standard use of flow matching) and also the GNN encoder for embedding initial distributions (this is novel). See supplementary PDF Algorithm 1 for training details.
>
> >How would this meta-learning framework compare with using the embedding from a pretrained encoder, such as a VGAE, or a classification model?
>
> This is an interesting direction, but was not the focus of our work. Our objective was to amortize learning over multiple distributions, so we can generalize to unseen distributions in the test set. We believe exploring pre-training of the distributional embedding encoder (training algorithms, losses) as well as exploring other architecture/models (VGAE, classification model) are natural and fruitful directions for future extensions of MFM. We leave this for future work and have discussed this in the text.
>
> >how much does the orientation-invariant embedding from the GNN help the model? Would any other encoder such as an MLP, still yield good performance?
>
> We use the permutation invariant property (not orientation-invariance) of the GNN model to learn embeddings for entire distributions. We explore the need for the GNN versus MLP by conducting an ablation over $k$. Specifically, we have included results for $k=0$, where the GNN encoder reduces to an MLP to try and deduce if using a GNN is necessary. We observe in our experiments that in some cases the MLP parameterization is sufficient for generalization across distributions, while in other settings $k>0$ performs better (see supplementary PDF).
>
> >How well does it scale to high dimensions?
>
> We have included additional high-dimensional experiments (no PCA used, dim=43) in the supplementary PDF (see general response). We observe that MFM performs consistently well in high dimensions as in the low dimensional settings.
>
> >How well does it scale to large datasets where you need to build large KNN graphs?
>
> The single-cell dataset we consider in this work has 2,500 pairs of marginals with 2k-13k cells for each source and target distribution pair. In our experiments, we show that we can efficiently train models for $k=100$ on this dataset. Training time increases with increasing $k$, but the performance gains saturate at certain values of $k$ (see updated experiments in supplementary PDF).
>
> >Would it scale effectively to high-dimensional tasks such as image generation?
>
> While our method can perform the conditional generation (e.g., class-conditional generation of images, see updated results in the supplementary PDF) its main purpose is to condition the dynamics on the starting population, i.e. the conditional variable is the entire collection of datapoints from the initial distribution (which we embed via a GNN). We are not aware of a suitable application of our method for image generation, but we remain open to any suggestions.
>
> >“We use CGFM to assess if other models are fitting the data.” Is this used as an evaluation metric in the experiments?
>
> CGFM is not an evaluation metrics but a baseline model that encodes source populations as one-hot-encoding vectors. CGFM indicates when the model can fit the training data and demonstrates that the test data is substantially different and cannot be generated by the same model (obviously, it does not correspond to any of the encodings). We have clarified this in the text.
>
>
> [1] Amos, Brandon, et al. "Meta optimal transport." ICML (2022).

---

> > ### Comment · Reviewer_BiEg · 2024-08-13
> >
> > I appreciate the authors' efforts in conducting ablation experiments and addressing the clarification questions.
> > The additional experiments have resolved my concerns in scalability and the necessity of a GNN.
> > I agree with the authors' clarifications (esp. with the algorithm box) on the novelty of joint training.
> > Therefore, I have raised my score.

---

### Official Review · Reviewer_HpVg · 2024-07-10

**Soundness:** 3
**Presentation:** 2
**Contribution:** 2
**Rating:** 5
**Confidence:** 4

**Summary:**

This paper proposed an extension of the Conditional Generative Modeling via Flow Matching (CGFM) framework. Taken inspiration from the theory of Wasserstein Gradient Flow, this new framework, Meta Flow Matching, proposed to learn the push-forward mapping of multiple measures in the same measures-space. This is motivated by the realistic problem of modeling single-cell perturbation data where we want to see the response of populations of cells of patients when receiving different treatments. A novelty of Meta Flow Matching is that by combining amortized optimization and CGFM, the trained MFM velocity network can model newly observed populations _without_ knowing their labels/conditions. Two empirical benchmarks were performed to showcase the effectiveness of MFM compared to the Flow Matching (FM) and CGFM.

**Strengths:**

- The method proposed is novel enough, and the problem is well-motivated. I also find the idea of integration of GNN to model the conditional variable quite neat. The method is based on the well-studied theory of Wasserstein gradient flow on measure spaces and amortized optimization framework.

- The empirical benchmark, especially on real biological data, seems to showcase the strength of MFM.

- Overall the paper is quite well written and easy to follow.

**Weaknesses:**

- The first part of the methodology section seems to be phrased as a new methodological contribution, but if I'm not mistaken this is just more or less restating the already established theory of W2 gradient flow and continuity equation (eq 14). I think the authors should put Section 3.1 into the background section (2nd Section).

- There is a lack of discussion on whether the 3 crucial assumptions (line 145-152) are satisfied in a realistic biological setting. For example, in theory, Assumption (iii) on the unique existence of the Cauchy problem stands when the velocity field satisfies some regularity conditions -- I'm not sure this can be extended to a parameterized neural network that takes input from another (graph) neural network as an embedding function, which is hardly Lipschitz smooth in most of the case.

- Algorithm boxes at the end of section 3 is highly welcome. Or if the authors cannot allocate the space, I highly recommend putting two (one for training and one for sampling) into the Appendix. It is quite hard to follow how the velocity is trained in reality. For example, what function $f_t(x_0, x_1)$ did the authors take for this work? Is it still linear interpolation as vanilla flow matching? Or does it involve adding some form of stochasticity as in stochastic interpolant or VP-SDE as in diffusion model? Is the coupling $(x_0, x_1)$ sampled to match randomly, or they are sampled to some form of alignment as in the multisample flow matching paper (Pooladian et al. 2023)?

- This might not be the original purpose of this work, but I would love to see how MFM perform on conditional image generation task. One can pick a simple small dataset such as CIFAR10 that already includes class labels, or better yet ImageNet dataset. The performance in this takse will be much more convincing than the synthetic experiment, where I would argue would target the same type of task.

**Questions:**

See weaknesses.

**Limitations:**

See weaknesses.

---

> ### Author Rebuttal · Authors · 2024-08-07
>
> >The first part of the methodology section seems to be phrased as a new methodological contribution, but if I'm not mistaken this is just more or less restating the already established theory of W2 gradient flow and continuity equation (eq 14). I think the authors should put Section 3.1 into the background section (2nd Section).
>
> Section 3.1 discusses ODEs rather than gradient flows on the W2 manifold, which is equivalent only if the vector field is defined as the W2 gradient of some functional. For instance, diffusion and the porous medium equation are gradient flows [1] but the Schrödinger equation is not [2]. The role of this section is to motivate the formalism used with examples. We have clarified this in the main body of the text.
>
> > There is a lack of discussion on whether the 3 crucial assumptions (line 145-152) are satisfied in a realistic biological setting. For example, in theory, Assumption (iii) on the unique existence of the Cauchy problem stands when the velocity field satisfies some regularity conditions -- I'm not sure this can be extended to a parameterized neural network that takes input from another (graph) neural network as an embedding function, which is hardly Lipschitz smooth in most of the case.
>
> In general, the considered ODEs on the W2 manifold correspond to PDEs on the state space, and the existence and uniqueness of their solution requires an extensive study in every particular case (e.g., see [3]). In practice, we find it rather unrealistic to recover the ground true PDE and the necessary assumptions on the vector field based solely on samples from the marginals as considered in Section 5. Although an interesting direction for future research, this is beyond the scope of our work.
>
> > Algorithm boxes at the end of section 3 is highly welcome. Or if the authors cannot allocate the space, I highly recommend putting two (one for training and one for sampling) into the Appendix ... For example, what function $f(x_0, x_1)$ did the authors take for this work? Is it still linear interpolation as vanilla flow matching? Or does it involve adding some form of stochasticity as in stochastic interpolant or VP-SDE as in diffusion model? Is the coupling $(x_0, x_1)$ sampled to match randomly, or they are sampled to some form of alignment as in the multisample flow matching paper (Pooladian et al. 2023)?
>
> Thank you for your suggestion! We have added the algorithm pseudocode into the main body of the final version of the paper (see the supplementary PDF Algorithm 1 for the pseudocode) and clarified the details in the text further (e.g., that we use independent samples from the marginals and linear interpolation as in the standard flow matching). Also, all the details of the algorithm can be found in the code supplemented to the submission.
>
> Sampling is accomplished by taking trained models $v_t(\cdot;\omega^*)$, $\varphi(\cdot;\theta^*)$, and given an input initial population, first compute population embeddings $h = \varphi\left(\{x_0^j\}_{j=1}^{N'};\theta\right)$, then solve $x_1^j = \int_0^1 v_t(x_t^j | h,c;\omega)^{2} dt + x_0^j$ via ODE solver. We have added this as Algorithm 2 in the main text.
>
> > This might not be the original purpose of this work, but I would love to see how MFM perform on conditional image generation task. One can pick a simple small dataset such as CIFAR10 that already includes class labels, or better yet ImageNet dataset. The performance in this takse will be much more convincing than the synthetic experiment, where I would argue would target the same type of task.
>
> While our method can perform the conditional generation (e.g., class-conditional generation of images, see updated results in the supplementary PDF) its main purpose is to condition the dynamics on the starting population, i.e. the conditional variable is the entire collection of datapoints from the initial distribution (which we embed via a Graph Neural Network). Hitherto, we are not aware of a suitable application of our method for image generation, but we remain open to any suggestions.
>
> [1] Otto, Felix. "The geometry of dissipative evolution equations: the porous medium equation." (2001): 101-174.
>
> [2] Chow, Shui-Nee, Wuchen Li, and Haomin Zhou. "Wasserstein hamiltonian flows." Journal of Differential Equations 268, no. 3 (2020): 1205-1219.
>
> [3] Schaeffer, Jack. "Global existence of smooth solutions to the Vlasov Poisson system in three dimensions." Communications in partial differential equations 16, no. 8-9 (1991): 1313-1335.

---

> > ### Comment · Reviewer_HpVg · 2024-08-12
> > **Thank you for the clarification**
> >
> > I have read the rebuttal of the authors. I remain in my opinion that the background part in Section 3.1 should be moved back to the earlier section to not be confused as a contribution. I also understand that it is quite hard to find theoretical analysis for this type of work (that leans more on methodological and algorithmic contributions). While the authors' rebuttal clarifies plenty of my concerns, I think the work still requires some reorganization and additional modification. Therefore, I keep my Borderline Acceptance score.

---

> > > ### Author Response · Authors · 2024-08-12
> > >
> > > Thank you for your time and effort in reading and responding to our rebuttal. Following your suggestion, we have moved the content of Section 3.1 to the background section to avoid any possible confusion and to make our contribution clearer.
> > >
> > > We are happy to consider any additional suggestions and to make further changes to improve the overall quality of our work.

---

### Official Review · Reviewer_mrP3 · 2024-07-10

**Soundness:** 3
**Presentation:** 3
**Contribution:** 3
**Rating:** 6
**Confidence:** 4

**Summary:**

The paper discussed the novel problem setup of generative modeling of the dynamics of probability distributions. The paper proposed Meta Flow Matching (MFM), an extension of the flow matching framework for implicitly learning the vector fields on the Wasserstein manifold of probability distributions. The paper demonstrated better transferability of the proposed framework on unseen distributions on both synthetic datasets and real-world drug-screen datasets.

**Strengths:**

- The problem setup of learning a flow matching model for mappings between distributions (i.e., a probability path on the Wasserstein manifold), to the best of my knowledge, is novel and has not been explored in previous work.
- The idea of using distribution-specific embeddings (the population embeddings) is well explained and motivated in the paper.
- The proposed method demonstrates better transferability on both synthetic and real-world datasets compared to other baselines.

**Weaknesses:**

- The proposed method seems to be a special case of a conditionally trained flow matching model where the conditions are continuous learnable embeddings. Such an idea has already been applied in various diffusion or flow matching models including image generation (conditioned on text embedding in the latent space), protein co-design [1] (conditioned on sequence, generate protein structure, or vice versa), and peptide design [2] (conditioned on receptor proteins, generate peptides).

- The idea of population embedding in the paper is similar to task embedding, which has been well-explored in the meta learning (e.g. [3]). Although the authors claimed their framework to be *meta* flow matching, related work in meta learning seems to lack.


[1] Campbell, Andrew, et al. "Generative flows on discrete state-spaces: Enabling multimodal flows with applications to protein co-design." arXiv preprint arXiv:2402.04997 (2024).

[2] Li, Jiahan, et al. "Full-Atom Peptide Design based on Multi-modal Flow Matching." arXiv preprint arXiv:2406.00735 (2024).

[3] Achille, Alessandro, et al. "Task2vec: Task embedding for meta-learning." Proceedings of the IEEE/CVF international conference on computer vision. 2019.

**Questions:**

1. What is the major difference between the proposed method and a conditionally trained flow matching model? See Weakness 1 for some examples of conditional generative models in other fields. These models also rely on continuous learnable embeddings as conditions during training and sampling.

2. If MFM can recovers the conditional generation via flow matching (CGFM) (Proposition 1), what is the benefit of using the proposed scheme over the latter? Can you provide more details regarding training CGFM? For example, what are fed into the model as conditions for CGFM?

3. There is a (minor) mismatch between Figure 1 and the MFM objective in Eq.15. In Figure 1 (or Eq.14), the flow matching model operates on the probability distributions to output a vector field $v\_t(p\_t)$ in the tangent space (an affine subspace as the probabilities needs to be normalized). However, in Eq.15, the flow matching model still operates on the data space $v\_t(x\_t)$. This is probably because the probability distributions are only implicitly described via data samples. Nonetheless, the authors should avoid saying in the caption of Figure 1 that the model learns a vector field on the Wasserstein manifold. During both training and sampling, the learned vector field is always defined on the data space in this work.

4. GNNs is a reason choice for data with geometric properties. However, the single-cell data does not seem to a exhibit a simple Euclidean geometry for GNN to work. Can you provide more justifications? Is there a better choice for the population embedding?

**Limitations:**

The authors have adequately and properly discussed the limitations and potential societal impact of the paper in the Appendix.

---

> ### Author Rebuttal · Authors · 2024-08-07
>
> >Such an idea has already been applied in various diffusion or flow matching models including image generation, protein co-design [1], and peptide design [2].
>
> The reviewer is correct in that MFM is a conditionally trained flow matching model. Indeed, there are many conditionally trained flow matching models, and there will very likely be more in the future. We note that MFM is **more general** than existing work which considers vector conditionals. MFM expands this to conditioning on input dependent **graphs**. See highlighted $h^i(\theta)$ in Algorithm 1 (supplementary PDF). Existing methods such as MultiFlow [1] have conditions which are input and space independent. We note that Pepflow [2] is concurrent work as it was not made publicly available until after the submission deadline (June 2).
>
> >What is the major difference between the proposed method and a conditionally trained flow matching model? These models also rely on continuous learnable embeddings as conditions during training and sampling.
>
> We have added an explanation of how MFM differs from the existing methods by including a discussion of the previous point/question. To be precise here, MFM depends on **distributions** of vectors instead of (potentially learned) vector conditionals, listed in Equations (14) and (15). To improve clarity, we have added details regarding the training of MFM (Algorithm 1 in the supplementary PDF) and a high level explanatory figure (Figure 1 in the supplementary PDF).
>
> >The idea of population embedding in the paper is similar to task embedding, which has been well-explored in the meta learning (e.g. [3]). Although the authors claimed their framework to be meta flow matching, related work in meta learning seems to lack.
>
> We thank the reviewer for bringing up this interesting connection to the field of meta learning. We have added a section on other "meta" — in the sense of amortization — frameworks, in our related works. Our work follows the naming convention of *Meta Optimal Transport* [4], and is "meta" in the same sense of amortizing OT or flow problems over multiple input distributions. This is related to how "meta" is used in the meta learning setting (which amortizes over learning problems).
>
> Here is a quick summary of the differences between our setting and Task2Vec [3]:
> 1. Task2Vec uses a task embedding in $R^d$ over datasets of images to condition a ML model for meta learning while we learn a task embedding over an input point cloud to condition a meta flow model.
> 2. Task2Vec proposes to use the (diagonal of the) Fisher information metric (FIM) of the dataset as the task embedding while we propose to learn a GNN that outputs an embedding for the downstream task. In other words, we learn our task embedding for the end-to-end performance while Task2Vec takes the FIM as the task conditioning information. In the spirit of Task2Vec, we will clarify that we could consider other embeddings of the point clouds, which could be related to their FIM, or other statistics of them — the FIM and other task embeddings may still be nearly computationally intractable for our larger point clouds, so we defer these ablations to future work as they may not be straightforward and may involve some tradeoffs.
>
> >If MFM can recovers the conditional generation via flow matching (CGFM) (Proposition 1), what is the benefit of using the proposed scheme over the latter? Can you provide more details regarding training CGFM? For example, what are fed into the model as conditions for CGFM?
>
> The benefit of MFM is that it can generalize to unseen populations where CGFM cannot. One-hot conditions for initial populations (and also treatments) are fed into CGFM. CGFM does not see the one-hot conditions for the initial populations in test sets. Hence, CGFM cannot generalize since it uses one-hot conditions and does not learn representations of the initial populations. We state this in lines 301-310.
>
> >In Figure 1 (or Eq.14), the flow matching model operates on the probability distributions to output a vector field $v_t(p_t)$ in the tangent space ... However, in Eq.15, the flow matching model still operates on the data space $v_t(x_t)$. Nonetheless, the authors should avoid saying in the caption of Figure 1 that the model learns a vector field on the Wasserstein manifold. During both training and sampling, the learned vector field is always defined on the data space in this work.
>
> The reviewer is correct that we directly parameterize a model which operates on the data space, and that this implicitly defines a vector field on the Wasserstein manifold. We have clarified this difference in the caption of Figure 1.
>
> >GNNs is a reason choice for data with geometric properties. However, the single-cell data does not seem to a exhibit a simple Euclidean geometry for GNN to work. Can you provide more justifications? Is there a better choice for the population embedding?
>
> Single-cell data is most commonly modeled using K-nearest-neighbor graphs [5]. With many other methods building on top of this representation for [Visualization UMAP, Imputation MAGIC, batch correction MNN, and other tasks]. Even though it does not exhibit a simple Euclidean geometry, KNN-GNNs are still widely applicable in this domain. Improved models for population embeddings is an interesting direction, which we leave to future work.
>
> [1] Campbell, Andrew, et al. "Generative flows on discrete state-spaces: Enabling multimodal flows with applications to protein co-design." (2024).
>
> [2] Li, Jiahan, et al. "Full-Atom Peptide Design based on Multi-modal Flow Matching." (2024).
>
> [3] Achille, Alessandro, et al. "Task2vec: Task embedding for meta-learning." IEEE/CVF. (2019).
>
> [4] Amos, Brandon, et al. "Meta optimal transport." ICML (2022).
>
> [5] Heumos, L., Schaar, A.C., Lance, C. et al. Best practices for single-cell analysis across modalities. Nature Reviews Genetics. (2023).

---

> > ### Comment · Reviewer_mrP3 · 2024-08-12
> >
> > I thank the authors for providing more explanations and clarifications regarding the difference between the proposed method of MFM versus existing methods of CGFM and meta-learning. After clarifying the difference from existing work of conditional flow matching and task embedding in meta-learning, I believe this work indeed offers an alternative perspective that can be interesting to the community. In this regard, I raise my score from 5 to 6. I would suggest the authors highlight these distinctions in the revised manuscript.

---

### Official Review · Reviewer_btyA · 2024-07-13

**Soundness:** 3
**Presentation:** 3
**Contribution:** 2
**Rating:** 5
**Confidence:** 4

**Summary:**

This paper introduces Meta Flow Matching (MFM), a flow matching framework modeling interacting samples evolving over time by integrating vector fields on the Wasserstein manifold. The authors leverage a Graph Neural Network to embed populations of samples and thus generalize the method over different initial distributions. The authors demonstrate the method on individual treatment responses predictions on a large multi-patient single-cell drug screen dataset.

**Strengths:**

Novelty: The method uniquely considers population interactions, unlike previous flow matching methods that model samples individually.

Generalization: The authors extended conditioning on latent variables to conditioning on population index in section 3.2. The proposition in section 3.2 demonstrates that conditional flow matching can fit well within the MFM framework. The experiments show that MFM can generalize to unseen data, outperforming other methods in this regard.

**Weaknesses:**

In Table 1 of the synthetic experiment, the authors compared the performance of FM, CGFM and MFM of k=0,1,10,50. MFM doesn't seem to beat existing methods on the metrics and from the visualizations, it's hard to tell MFM is actually doing better than FM. Also, for the various values of k, some explanations on how performance correlates with values of k and why might be necessary for readers to understand this table.

In both experiments, the authors only compared FM, CGFM, and in Table 2 also ICNN. Probably more methods, like diffusion, should also be taken into comparison. Also, in experiment 2, only W1, W2 and MMD are computed as metrics. While these are useful when modeling distributions, more metrics, especially those specific to this application may be applied.

**Questions:**

In Table 1, the authors compared k=0,1,5,10,50 and in Table2, only k = 0 and 10 are listed. Some explanations on how this decision is made is probably helpful.

**Limitations:**

The authors have not addressed limitations.

---

> ### Author Rebuttal · Authors · 2024-08-07
>
> >In Table 1 of the synthetic experiment ... MFM doesn't seem to beat existing methods on the metrics and from the visualizations, it's hard to tell MFM is actually doing better than FM.
>
> We thank the reviewer for the opportunity to further clarify our results. Metric-wise from Table 1, MFM outperforms FM and CGFM yielding lower W1 and W2 metrics on the test data (Y's). Moreover, visually FM fails to de-noise the letters in any way on both the train and test sets, while MFM is much closer to achieving the target distributions. We have updated the synthetic experiment (see Table 3 in the supplementary PDF), where we have made the task harder (test letters are completely unseen during training, i.e. in any orientation). Here it is again clear that MFM outperforms FM and CGFM across all metrics for the test set of Y's.
>
> >explanations on how performance correlates with values of k and why might be necessary for readers to understand this table.
>
> We thank the reviewer for pointing this out. We have added an explanation on how $k$ affects performance in the respective experiments sections. In these experiments we considered various values of $k$ primarily as an ablation to observe how performance changes for different $k$'s. We also wanted to observe the role/importance of considering interactions between "particles" (or samples) for learning population embeddings. For example, when $k > 0$ particle interactions are incorporated via a knn graph, while for $k=0$ (DeepSets, permutation invariant MLP) no particle interactions are incorporated. We found:
> - **Synthetic experiments:** higher $k$ correlates with better performance on the synthetic experiments.
> - **Single-cell experiments:** No clear single selection for $k$ that yields the best performance across all tasks on the single-cell experiments. We observe that for the _replicates_ split $k=0$ on average performs better than $k>0$. Whereas on the _patients_ split, the opposite is true.
>
> This is possibly since the _patients_ split forms a more difficult generalization problem (more diversity between training populations and test populations), and hence it is more difficult to over-fit during training with higher $k$.
>
>
> >the authors only compared FM, CGFM, and in Table 2 also ICNN. Probably more methods, like diffusion, should also be taken into comparison.
>
> The central focus of this work was to devise a general framework for learning population dynamics while conditioning on entire distributions — to generalize across previously unseen environments/distributions. We focused our work on  flow matching due to its versatility and training efficiency (relative to other generative modeling methods used in similar tasks) [1, 2, 3]. Moreover, flow matching has been shown to consistently yield competitive performance in single-cell population dynamics prediction tasks while also being easily extendable to incorporate optimal transport and stochastic formulations [4, 5, ]. **The performance trend witnessed for our flow matching models (MFM > FM, CGFM) should be agnostic to the choice of the backbone model. To show this, we have added an additional set of baseline models to all experiments that are akin to diffusion models** (see supplementary PDF Tables 1, 2, 3). We train FM$^{\text{w/}}\mathcal{N}$, CGFM$^{\text{w/}}\mathcal{N}$, and MFM$^{\text{w/}}\mathcal{N}$ which use a Gaussian source distribution for sampling, i.e. $x_0 \sim \mathcal{N}(0, 1)$, while $p_1$ remains the same set of target letter distributions. Here, MFM still uses the original $p_0$ in the GNN model to learn population embeddings, while the flow model uses $x_0 \sim \mathcal{N}(0, 1)$ as $p_0$. From the updated Tables, it is clear that (MFM$^{\text{w/}}\mathcal{N}$ > FM$^{\text{w/}}\mathcal{N}$, CGFM$^{\text{w/}}\mathcal{N}$), which is consistent with our expectation.
>
> >in experiment 2, only W1, W2 and MMD are computed as metrics. While these are useful when modeling distributions, more metrics, especially those specific to this application may be applied.
>
> Our choice of metrics is directly taken from the standard accepted in the community of single-cell population dynamic prediction and perturbation response prediction [4, 5, 6, 7]. For the single-cell experiments, we have added an additional metric used in [7]: the average correlation coefficient of the feature (bio-marker) means, labeled $r^2$. We have evaluated all single-cell models using this additional metric and observed MFM outperforms the baselines on $r^2$ as well (see supplementary PDF Tables 1 and 2).
>
> >In Table 1, the authors compared k=0,1,5,10,50 and in Table2, only k = 0 and 10 are listed. Some explanations on how this decision is made is probably helpful.
>
> Thank you for pointing this out. We have added a more comprehensive ablation over $k$ for the single-cell experiments (see supplementary PDF Tables 1 and 2). Specifically, we have trained MFM models for $k=0,10,50,100$ on the single-cell experiments.
>
> >The authors have not addressed limitations.
>
> We would like to clarify that we have included a discussion of limitations and broader impacts in the Appendix D and E.
>
> [1] Lipman, Yaron, et al. "Flow Matching for Generative Modeling." ICLR. (2023).
>
> [2] Albergo, Michael Samuel, and Eric Vanden-Eijnden. "Building Normalizing Flows with Stochastic Interpolants." ICLR. (2023).
>
> [3] Liu, Xingchao, and Chengyue Gong. "Flow Straight and Fast: Learning to Generate and Transfer Data with Rectified Flow." ICLR. (2023).
>
> [4] Tong, Alexander, et al. "Improving and generalizing flow-based generative models with minibatch optimal transport." TMLR. (2024).
>
> [5] Tong, Alexander, et al. "Simulation-free schrodinger bridges via score and flow matching." AISTATS. (2024).
>
> [6] Neklyudov, Kirill, et al. "A computational framework for solving Wasserstein Lagrangian flows." ICML. (2024).
>
> [7] Bunne, C., Stark, S.G., Gut, G. et al. Learning single-cell perturbation responses using neural optimal transport. Nature Methods. (2023).

---

> > ### Comment · Reviewer_btyA · 2024-08-13
> >
> > I thank the authors for their response and raise my score from 4 to 5.

---

### Author Rebuttal · Authors · 2024-08-06

## Feedback Summary

We thank all the reviewers for the time they invested in reviewing our paper and for their valuable and constructive feedback that will help improve our work's overall quality.

In this work, we introduced **Meta Flow Matching (MFM)**, a novel framework for learning the dynamic evolution of populations with the objective to generalize across previously unseen populations/micro-environments. By amortizing the flow model over initial populations -- using a GNN architecture to learn conditional embeddings of entire populations -- **we show that MFM can successfully perform prediction on previously unseen test populations in synthetic and real experiments, where baseline methods fail.**

In general, there is consensus among all reviewers that **MFM** is a **novel** and **unique** framework with convincing empirical experiments that showcase its **significance** and ability to generalize across previously unseen populations. In addition, reviewers found our work to be well motivated [mrP3, HpVg] with clear exposition and presentation [mrP3, HpVg, BiEg]. The primary concerns brought up by the reviewers were questions regarding clarifications and experiments. Below we outline how we have addressed these items.

**Improved Experiments:** reviewers asked about the scalability of MFM to larger dimensions and data sets [HpVg, BiEg], ablations over the GNN architecture [btyA, BiEg], and consideration of additional baselines and metric(s) [btyA]. To address these items, we have improved the synthetic and real-data empirical experiments and included the updates in the **supplementary PDF**.
- [HpVg, BiEg] generally asked about considering larger scale and higher dimensional experiments. To address this, we **added high-dimensional experiments** (see Tables 1 and 2 in the supplementary PDF) on the full organoid-drug screen dataset (no PCA reduction used) and demonstrated that MFM outperforms all baselines across all tasks and metrics.
- [btyA] asked about considering application-specific metrics for the experiments on single-cell perturbation data. To address this, we **added the squared correlation metric ($r^2$)** (see Tables 1 and 2 in the supplementary PDF) commonly used in single-cell applications [1]. MFM consistently outperforms baselines on $r^2$.
- [btyA, BiEg] asked about further ablations over the GNN population embedding architecture. To address this, we have **added a more comprehensive ablation over the nearest-neighbor parameter $k$** (see Tables 1 and 2 in the supplementary PDF) on the single-cell data experiments ($k=0,10,50,100$).
- [btyA] asked about additional baselines while [HpVg, BiEg] asked about conditional generation tasks. To address this, we have **added additional baseline models akin to diffusion** (see Tables 1, 2, and 3 in the supplementary PDF) where we use $\mathcal{N}(0, 1)$ as the source distribution. We trained and evaluated these additional baselines on our synthetic and single-cell experiments, showing that MFM also outperforms baseline models in this regime.
- [btyA] had concerns regarding the performance of MFM on the synthetic experiments. To address this, we have **improved MFM by adding _batching over initial population_ during training** (see supplementary PDF, Algorithm 1 where we added the line $i \sim \mathcal{U}_{\{1,N\}}(i)$). In addition, we have also made the overall synthetic letters task more difficult. Originally, 1 orientation of the test letters (Y's) was seen during training. In the updated experiments, the test letter populations are **not** seen in any orientation -- i.e. Y's are entirely unseen during training. We demonstrate that MFM outperforms baselines across all metrics in this experiment.

**Clarifications:** In general, the reviewers asked clarifying questions about the high level details of MFM and to expand on the training procedure of MFM [mrP3, HpVg]. To address these questions, we have **added Algorithm 1** outlining the training procedure for MFM (and similarly CGFM and FM). Furthermore, we have **added an additional explanatory figure** illustrating and comparing the central high-level elements of FM, CGFM, and MFM of our work. Additionally, reviewers asked questions regarding differences between MFM, CGFM, and existing/concurrent approaches [mrP3, HpVg, BiEg]. We address all of these concerns in individual responses and include any relevant additions in the **supplementary PDF**.

We believe we have addressed all the concerns and questions posed by the reviewers, improving the overall quality of our paper. We once again thank all reviewers for their insightful feedback and hope they will consider raising their scores considering the numerous additions we have made to improve the clarity, impact, and significance of our work.

[1] Bunne, C., Stark, S.G., Gut, G. et al. Learning single-cell perturbation responses using neural optimal transport. Nature Methods. (2023).

---

### Decision · Program_Chairs · 2024-09-25

**Decision:**

Reject

**Comment:**

This paper proposes an innovative approach to embedding the initial distribution in Flow Matching problems, using a GNN to learn embeddings that generalize to unseen initial distributions. While the idea shows promise, the overall enthusiasm among reviewers was moderate, with concerns about the significance of the results and scalability. Although the authors addressed many of these concerns in their response, the reviewers remained cautious. Reviewer BiEg provided a strong score of 7 and could have championed the paper, but other reviewers maintained scores ranging from 5 to 6. Given the mixed reviews, I recommend rejecting the paper at this time.